# The Predictive Accuracy of the General Movement Assessment for Cerebral Palsy: A Prospective, Observational Study of High-Risk Infants in a Clinical Follow-Up Setting

**DOI:** 10.3390/jcm8111790

**Published:** 2019-10-25

**Authors:** Ragnhild Støen, Lynn Boswell, Raye-Ann de Regnier, Toril Fjørtoft, Deborah Gaebler-Spira, Espen Ihlen, Cathrine Labori, Marianne Loennecken, Michael Msall, Unn Inger Möinichen, Colleen Peyton, Annamarie Russow, Michael D. Schreiber, Inger Elisabeth Silberg, Nils Thomas Songstad, Randi Vågen, Gunn Kristin Øberg, Lars Adde

**Affiliations:** 1Department of Neonatology, St. Olavs hospital, Trondheim University Hospital, 7006 Trondheim, Norway; 2Department of Clinical and Molecular Medicine, Norwegian University of Science and Technology, 7491 Trondheim, Norway; Toril.Fjortoft@stolav.no (T.F.); lars.adde@ntnu.no (L.A.); 3Ann and Robert H Lurie Children’s Hospital of Chicago, Chicago, IL 60611, USA; LBoswell@luriechildrens.org (L.B.); r-deregnier@northwestern.edu (R.-A.d.R.); ARussow@luriechildrens.org (A.R.); 4Feinberg School of Medicine, Northwestern University, Chicago, IL 60611, USA; dgaebler@sralab.org (D.G.-S.); colleen.peyton1@northwestern.edu (C.P.); 5Clinic of Clinical Services, St. Olavs hospital, Trondheim University Hospital, 7006 Trondheim, Norway; randi.tynes.vagen@stolav.no; 6Shirley Ryan AbilityLab, Chicago, IL 60611, USA; 7Department of Neuromedicine and Movement Science, Norwegian University of Science and Technology, 7491 Trondheim, Norway; espen.ihlen@ntnu.no; 8Department of Clinical Therapeutic Services, University Hospital of North Norway, 9038 Tromsø, Norway; Cathrine.Labori@unn.no (C.L.);; 9Department of Pediatrics, Division of Paediatric and Adolescent Medicine, Oslo University Hospital, 0372 Oslo, Norway; Marianne.loennecken@gmail.com (M.L.); umoinich@ous-hf.no (U.I.M.); isilberg@ous-hf.no (I.E.S.); 10University of Chicago Medicine, Comer Children’s Hospital, Section of Developmental and Behavioral Pediatrics, Chicago, IL 60637, USA; mmsall@peds.bsd.uchicago.edu; 11University of Chicago Kennedy Research Center on Intellectual and Neurodevelopmental Disabilities, Chicago, IL 60637, USA; 12University of Chicago Medicine, Comer Children’s Hospital, Department of Pediatrics, Chicago, IL 60637, USA; mschreiber@peds.bsd.uchicago.edu; 13Department of Pediatrics and Adolescent Medicine, University Hospital of North Norway, 9038 Tromsø, Norway; Nils.Thomas.Songstad@unn.no; 14Department of Health and Care Sciences, Faculty of Health Sciences, UiT- The Arctic University of Norway, 9019 Tromsø, Norway

**Keywords:** general movement assessment, fidgety movements, cerebral palsy, early brain damage, neuroimaging

## Abstract

Background: Early prediction of cerebral palsy (CP) using the General Movement Assessment (GMA) during the fidgety movements (FM) period has been recommended as standard of care in high-risk infants. The aim of this study was to determine the accuracy of GMA, alone or in combination with neonatal imaging, in predicting cerebral palsy (CP). Methods: Infants with increased risk of perinatal brain injury were prospectively enrolled from 2009–2014 in this multi-center, observational study. FM were classified by two certified GMA observers blinded to the clinical history. Abnormal GMA was defined as absent or sporadic FM. CP-status was determined by clinicians unaware of GMA results. Results: Of 450 infants enrolled, 405 had scorable video and follow-up data until at least 18–24 months. CP was confirmed in 42 (10.4%) children at mean age 3 years 1 month. Sensitivity, specificity, positive and negative predictive values, and accuracy of absent/sporadic FM for CP were 76.2, 82.4, 33.3, 96.8, and 81.7%, respectively. Only three (8.1%) of 37 infants with sporadic FM developed CP. The highest accuracy (95.3%) was achieved by a combination of absent FM and abnormal neonatal imaging. Conclusion: In infants with a broad range of neonatal risk factors, accuracy of early CP prediction was lower for GMA than previously reported but increased when combined with neonatal imaging. Sporadic FM did not predict CP in this study.

## 1. Introduction

Cerebral palsy (CP) is the major cause of motor impairment in young children [1,2], and the majority of individuals live with associated impairments. As there is no specific diagnostic test for CP, making a diagnosis depends on a set of criteria including disorders in tone and posture [1]. The clinical picture changes over time [3], and a recent review showed that referral for diagnosis typically happens between 10 and 21 months of age [4]. In addition, reconfirmation of the diagnosis between 3 and 5 years may be required [1]. The etiology of CP is multifactorial and heterogenous and is characterized by an injury to the immature brain [5]. Although many children who develop CP do not have identifiable risk factors during the neonatal period, there is reason to explore how clinicians can identify affected infants with perinatal risk factors earlier. Earlier identification may facilitate interventions during infancy when neuroplasticity is high [6], improve access to community services [7] and improve well-being for parents [8]. In accordance with this, a recently published clinical recommendation strongly encourages making a diagnosis of CP before 5 months corrected age in infants with newborn-detectable risk factors, abnormal cerebral magnetic resonance imaging (MRI), and neurological abnormalities [9].

The General Movement Assessment (GMA) during the fidgety movements (FM) period (2–4 months post-term age (PTA)) is currently considered the best clinical test for early prediction of CP [9,10], and several studies have shown high sensitivities and specificities [11,12,13,14,15,16]. However, there are few studies of cohorts with the variety of risk factors typically seen in neonatal intensive care follow-up clinics. Additionally, high sensitivities and specificities are often reported by GMA experts who are also tutors [12,13,14,15], or from populations with a high prevalence of CP [11,12].

Fidgety movements can be classified according to their presence and length of interspersed pauses [17]. Intermittent or continual FM are considered normal, absent, or sporadic abnormal, and exaggerated FM may have heterogenous outcomes [18]. The clinical significance of the different categories is, however, not clear. CP outcomes of infants with sporadic FM have, to the best of our knowledge, not been prospectively studied in large cohorts of high-risk infants. In order to provide accurate and balanced information to parents, clinicians need to know the predictive validity of the different categories of FM in clinically relevant populations.

In this study, the predictive accuracy of GMA assessed once during the FM period for later CP was explored in a large population of neonatal intensive care unit (NICU) graduates referred to neurodevelopmental follow-up at discharge. The effect of the temporal organization of FM on CP-risk and gross motor function was also explored, as well as how neonatal imaging contributed to the prediction of CP.

## 2. Material and Methods

High-risk infants were consecutively recruited before discharge from the NICU at five sites from 2009–2014. Sites 1–3 recruited infants in three Norwegian university hospitals based on at least one of the following: (a) Birth weight (BW) ≤1000 g and/or gestational age (GA) <28 (extremely low birth weight/extremely low GA; ELBW/ELGAN), (b) neonatal arterial ischemic stroke, (c) neonatal encephalopathy, (d) other significant risk factors for perinatal brain injury (see Table 1 for all included infants). Site 4 recruited infants in a quaternary NICU in Chicago, USA with at least one of the following: (a) GA < 29 weeks, (b) congenital heart disease (CHD) in need of cardiac surgery before 10 weeks of age, (c) medically complex infants including congenital anomalies and/or infants with chromosomal abnormalities with an extended NICU-stay beyond 10 weeks PTA, (d) infants admitted to the NICU due to neurological symptoms and abnormal neonatal brain imaging. Site 5 recruited infants in a tertiary NICU in Chicago, USA who were born before 31 weeks GA with a BW < 1500 g, required oxygen at birth and were enrolled in a randomized controlled trial of two different doses of inhaled nitric oxide for neuroprotection (NOVA2 trial; https://clinicaltrials.gov/ct2/show/NCT00515281). Although one infant could have several risk factors for adverse neurodevelopment, all infants were classified into one risk group according to their main reason for referral. Extremely preterm infants (ELBW/ELGAN) were classified as such irrespective of other risk factors, whereas preterm infants with GA 28^0^–30^6^ weeks and BW > 1000 g were classified as such only if they had no other risks of perinatal brain injury (e.g., moderate-to-severe imaging abnormalities).

Cerebral imaging (ultrasound (cUS) or MRI) was done for clinical purposes and according to each unit’s guideline. A central classification of the imaging results into normal or abnormal was based on local written reports, by an experienced neonatologist (RS) who was unaware of the infant’s medical history, GMA result, and neurodevelopmental outcome. Cerebral imaging was classified as abnormal if there was intraventricular hemorrhage (IVH) over grade II according to Papile [19], other significant intracranial hemorrhage with symptoms leading to admission to the NICU (usually seizures), cystic periventricular leukomalacia (cPVL), ventricular dilatation consistent with previous PVL, moderate-to-severe hypoxic-ischemic injury according to Rutherford [20], neonatal stroke, or sinus venous thrombosis with injury of underlying cerebral parenchyma. Milder abnormalities, not known to be associated with an increased risk of CP, were categorized as normal. A pediatric radiologist was consulted whenever there was doubt about the classification.

Infants were video recorded during active wakefulness at 10 to 15 weeks PTA using a standardized set-up with a commercially available digital video camera (Sanyo VPC-HD2000, SANYO Electric Co, Ltd., Osaka, Japan). Video recordings and gestalt assessment of GMs were done according to Prechtl and co-workers [18]. Two trained and certified GMA observers (LA and TF) who were blinded to the medical history of the infants performed all assessments. GMA was classified as abnormal if FM were absent or only sporadically present, and as normal if present intermittently or continually [17]. Seven infants with present FM which were excessive in amplitude and speed (exaggerated), none of whom developed CP, were a priori excluded from the analysis due to the unpredictable outcomes of this category. In case of disagreement about GMA classification, the observers re-assessed the video together and reached consensus. For infants who had more than one video recording taken during the FM period, the one closest to PTA 12 weeks was selected for analysis.

Children were followed clinically to establish the diagnosis of CP or no CP. Some children with CP were followed to the age of 6 years but follow-up to exclude CP continued until a minimum of 18–24 months at all sites.

CP and CP subtype were diagnosed as part of regular follow-up by trained pediatricians who were unaware of the GMA classification and in accordance with the decision-tree published by the Surveillance of cerebral palsy in Europe (SCPE) [1]. The use of the SCPE decision-tree for making a CP diagnosis was agreed upon by all sites prior to study start. Gross motor function was classified using the Gross motor function classification system [21].

Statistics: Data were analyzed using SPSS Statistics version 23.0 (IBM SPSS Statistics, Chicago, IL). The data are presented as numbers with proportion (%) or mean with standard deviation (SD). Group differences were examined using the chi-square test or Fisher exact test for dichotomous variables. Sensitivity, specificity, positive and negative predictive values, and test accuracy were estimated. Logistic regression was used to assess the association between the explanatory variables GMA (normal vs. abnormal), neonatal imaging (normal vs. abnormal), GA and sex, and the dichotomous outcome variable CP. Results from logistic regression are presented as odds ratios (OR) with 95% CI. Results were considered statistically significant for *p* values less than 0.05.

Ethical Approval: The study was approved by the regional committee for medical and health research ethics (REC Central-Committee 4.2007.2327) in Norway. Each of the two study sites in Chicago, USA also had approvals from their local Institutional Review Boards. Written parental consent was obtained before inclusion.

## 3. Results

A total of 450 infants were included. Infants with video recordings which could not be assessed due to inadequate video set-up (*n* = 5), infant state of crying/fussing (*n* = 8) or had external distractions during recording (*n* =2), and infants with restricted movements due traumatic injury of an extremity (*n* = 1) were excluded. In addition, infants who had significant medical incidences occurring after GMA assessment (*n* = 3), or who were categorized as having exaggerated FM (*n* = 7) or were lost to follow-up (*n* = 19) were excluded. These exclusions left 405 infants available for analysis. Demographic characteristics and the clinical condition considered the most relevant for increased risk of perinatal brain injury are shown in Table 1.

### 3.1. GMA and Neonatal Imaging

Mean age at video recording for GMA was 12.3 weeks (SD 1.3). Ninety-six (23.7%) of the infants were categorized with abnormal GMA (60 (14.8%) with absent and 36 (8.9%) with sporadic FM), and 309 (76.3%) infants were categorized as normal (258 (63.7%) with intermittent, 51 (12.6%) with continual FM).

Neonatal cUS and/or MRI results were available in all but two infants. In preterm infants, cUS was the most common modality with results available for all those born before GA 32 weeks. MRI results were available in 104 (68.4%) of those born at GA ≥ 32 weeks.

Eighty-seven (21.5%) infants had abnormal cerebral imaging. The proportion of infants with abnormal GMA during the FM period was significantly higher among those with abnormal compared to normal imaging (42.5% vs. 18.4%, respectively; *p* < 0.001).

### 3.2. Prediction of CP

Forty-two (10.4%) children received a diagnosis of CP, and mean age at the last available confirmation of the diagnosis was 3 years 1 months (SD 12.4 months, range 1.2–6 years). Of the 42 children with CP, eight (19%) had unilateral spastic, 26 (62%) had bilateral spastic, five (12%) had dyskinetic, and one (2%) child had ataxic CP. In two children data on CP subtype was not available.

Predictive accuracy of GMA and neonatal imaging is presented in Table 2. Twenty-nine (48.3%) of those with absent FM were diagnosed with CP, compared to three (8.1%) and 10 (3.9%) of those with sporadic and intermittent FM, respectively. No infant with continual FM developed CP. The highest overall accuracy of CP prediction was achieved by a combination of absent FM and abnormal imaging (95.3 % (95 % CI, 92.7–97.1); Table 2). This was mainly due to a very high specificity of 99.2% (95 % CI, 97.6–99.8)

The odds of developing CP were 14 times higher for infants with absent/sporadic FM in a model which included GA, sex, and neonatal imaging (Table 3).

Gross motor function classification (GMFC) was available for all children with a diagnosis of CP. Twenty-one (50%) had a GMFCS level of IV–V, corresponding to non-ambulatory function. There was a significant association between absence of FM and being non-ambulatory among those who developed CP. Nineteen (66%) of the 29 infants with absent FM who developed CP had a GMFCS level of IV–V compared to one (8%) of 13 with any presence (sporadically or intermittently) of FM (*p* < 0.001).

## 4. Discussion

In this study of a large and representative high-risk population of NICU graduates, sensitivity, and specificity of GMA during the FM period for later CP were lower than previously reported. Infants with sporadic FM appeared to have a low risk of CP, and they had better gross motor function if they developed CP compared to infants with absent FM.

Despite the long experience and demonstrated reliability [22] of the two certified GMA observers assessing all videos in this study, the test accuracy of GMA for later CP was lower than the above 90% sensitivities and specificities reported in several systematic reviews published over the last decade [10,23,24]. Many of the published studies on GMA have been conducted by researchers and GMA tutors [11,12,13,14,15,25,26,27] who are highly skilled experts with years of experience in developing and teaching the GMA. Those results may not be transferable to ordinary clinical settings with less experienced raters [28]. However, our results were very similar to what has been found in two single-center studies of preterm infants using local GMA raters from outside the highly skilled academic settings [28,29]. This may suggest that factors other than years of experience may influence the performance of the GMA. Experience from using GMA both in clinical work and research, as well as the opportunity to share experiences within an environment where the method is being taught and developed, may all play a role. Some studies also report test metrics based on serial assessments during the FM period [11,12,25,26,27]. We chose to include only one assessment, to preserve high retention rates and increase generalizability of results to clinical settings with restricted GMA resources. However, this may have contributed to lower accuracy compared to serial assessments providing developmental trajectories [30,31].

The present study included infants with a variety of clinical conditions frequently seen in tertiary and quaternary NICUs. Infants with severe congenital heart disease who needed early surgery [32] and infants with complex medical conditions requiring prolong stay in the NICU are rarely included in studies on GMA, and we argue that this makes our results robust and generalizable to a clinical setting. Some studies have included infants with a low or high risk of CP [11,12,25,26,33] or only those with cerebral imaging abnormalities [34]. Such selection bias may give an overrepresentation of infants who are relatively easy to classify and may not represent the accuracy of GMA in a clinical setting. We consider the 10% CP prevalence found in our study to be representative for many high-risk populations [35], and the predictive values which are reported are, therefore, relevant to clinicians. An Australian study of 259 high-risk infants included a patient population with a mix of risk factors similar to ours, but with a shorter follow-up period (12–24 months) and twice the CP prevalence as the present study [16]. By using a network of certified raters with a central rater settling any disagreement, they achieved a sensitivity of 98%. The Australian study did not use the category sporadic FM, but even when comparing only those with absent FM our study had substantial lower sensitivity (69% vs. 98%). Population differences and a slightly different classification system may have contributed to the differences in accuracy, and shorter follow-up with potential loss of milder CP phenotypes may overestimate the sensitivity reported by the Australian group.

Sporadic FM are considered abnormal when occurring between 9–16 weeks PTA [17,31], but the rate of CP among infants with sporadic FMs has not been specifically evaluated. In a study of 46 infants with abnormal general movements, only one of 10 infants with sporadic FM was diagnosed with CP [36]. Sporadic FM appears to be rare among typically developing infants [37] but had a prevalence of almost 9% in the present study. Although very few of them developed CP, other neurodevelopmental delays cannot be excluded based on the present study. Further studies exploring long-term outcomes of children with sporadic FM would be valuable to understand the relationship between temporal organization of FM and neurodevelopmental outcomes.

Neonatal imaging and GMA results only partly overlapped in the present study and considering both assessments increased the accuracy of early prediction. Classification of imaging results was not specifically targeted to detect the risk of CP, but more an overall evaluation of the severity of brain injury. Consequently, specificity might have increased with a more targeted classification. However, most cases classified with abnormal imaging had findings which are strongly associated with later CP (e.g., IVH grade III-IV and moderate-to-severe hypoxic-ischemic injury), which was supported by the high accuracy of abnormal imaging for later CP. Furthermore, the categorization of imaging results was based on local reports of imaging done for clinical purposes, making the results generalizable to clinical settings.

Among children with CP, those who presented with sporadic or intermittent FM, had significantly better gross motor function than those who presented with absent FM. A retrospective study of 61 children with CP concluded that sporadic FM did not indicate a milder type of CP [17]. However, also in that study the proportion of children with GMFCS 1 was higher among those with sporadic compared to absent FM (56% vs. 18%, respectively), but this difference was not statistically significant. Together with our findings, this suggests that there is a continuum of motor performance among children with CP which may be captured by the temporal organization of FM during infancy.

Successful implementation of “Early diagnosis and intervention guidelines for CP” has already been reported [38], and The Cerebral Palsy Foundation has taken initiative for a structured implementation of the guideline, which includes the GMA, across centers in the US (Cerebral Palsy Foundation, early recognition. Available at http://yourcpf.org/early-recognition). Byrne and co-workers [38] reported a significant decrease in age at CP diagnosis from pre- to post-implementation of the guidelines. However, the introduction of GMA did not seem to reduce the need for sequential motor assessments. Our results support that GMA during the FM period is excellent in identifying infants already at 2–4 months PTA who will not develop a severe form of CP. This knowledge may be used to target follow-up towards other domains of development in infants with present FM, as well as reassuring parents.

Limitations: A few children with CP in our study were assessed before two years of age for gross motor function. The GMFCS is more reliable after two years of age [3], and the GMFCS results must be interpreted with caution. Longer follow-up might also have resulted in more children presenting with a mild CP phenotype not identified during the study follow-up. In accordance with the strong association between presence of FM and milder types of CP, we speculate that this could have resulted in more infants with present FM being diagnosed with CP and, consequently, poorer performance of GMA.

The present study only focused on CP outcome of participants. Absent FM may be associated with genetic disorders and disabilities such as autism [39], which are likely to occur in a cohort like ours comprising infants with significant medical complexities and congenital abnormalities. However, the predictive accuracy of GMA for disabilities other than CP was beyond the scope of this study.

All study sites did not provide GMA results to parents, and the present study did not include any evaluation of parental reactions to early CP prediction. Although a delay in providing a diagnosis of CP is harmful to parents’ well-being [8], little is known about parental well-being when an infant is classified as having a high risk of CP or even a diagnosis of CP which later proves to be wrong. Future research should include the perspectives of parents of high-risk children both with and without CP.

## 5. Conclusions

Presence of FM was a strong marker for a non-CP outcome in high-risk infants but did not exclude milder CP phenotypes. The predictive accuracy of GMA increased when sporadic FM were not categorized as a marker for later CP, and absence of FM correctly predicted CP in approximately 50% of cases. Neonatal cerebral imaging in combination with GMA increased the predictive accuracy. Future research should focus on how neurodevelopmental follow-up could be individualized and targeted based on an early developmental profile. In infants with present FM, a redirection of resources to closer assessment of other areas of neurodevelopment, such as cognition, communication, and behavior, may be considered.

## Figures and Tables

**Table 1 jcm-08-01790-t001:** Demographic variables and primary reason for referral to follow-up.

Risk Groups	N (%)
BW ≤ 1000 g and/or GA < 28 weeks	188 (46.4)
Boys, n (%)	102 (54.3)
BW, mean (SD), grams	826 (183)
GA, mean (SD), weeks	26.2 (1.7)
BW > 1000 g and GA 28^0^–30^6^ weeks Neonatal arterial ischemic stroke	54 (13.1) 15 (3.6)
Neonatal encephalopathy	57 (13.8)
CHD w/surgery before 10 weeks	41 (10.1)
Others ^a^	50 (12.3)

BW, birth weight; GA, gestational age; CHD, congenital heart disease. ^a^ Others: Infants with significant abnormalities on cerebral imaging (intraventricular hemorrhage grade III–IV, other significant intracranial hemorrhages with or without seizures, cystic periventricular leukomalacia, ventriculomegaly, venous sinus thrombosis; *n* = 24), central nervous system infection (*n* = 6), medically complex infants (syndromes/chromosomal abnormalities, multiple congenital anomalies, hydrops fetalis, severe lung hypoplasia, protracted hypoglycemia, seizures with unknown etiology) (*n* = 16), and severe intrauterine growth restriction (*n* = 3). One second twin came to follow-up due to referral of the first twin.

**Table 2 jcm-08-01790-t002:** Predictive accuracy of the General Movement Assessment (GMA) and neonatal cerebral imaging for later cerebral palsy.

Predictive Accuracy/GMA and Imaging Results	Sensitivity % (CI 95%)	Specificity % (CI 95%)	PPV % (CI 95%)	NPV % (CI 95%)	Accuracy % (CI 95%)
Absent/sporadic FM *FN = 10, FP = 64*	76.2 (60.6–88.0)	82.4 (78.1–86.2)	33.3 (27.4–39.8)	96.8 (94.6–98.1)	81.7 (77.6–85.4)
Absent FM *FN = 13, FP = 31*	69.1 (52.9–82.4)	91.5 (88.1–94.1)	48.3 (38.7–58.1)	96.2 (94.2–97.6)	89.1 (85.7–92.0)
Abnormal neonatal imaging *FN = 8, FP = 53*	81.0 (65.9–91.4)	85.3 (81.2–88.8)	39.1 (32.5–46.1)	97.5 (95.4–98.6)	84.9 (81.0–88.2)
Absent/sporadic FM and/or abnormal imaging *FN = 5, FP = 107*	88.1 (74.4–96.0)	70.3 (65.3–75.0)	25.7 (22.3–29.6)	98.1 (95.7–99.1)	72.1 (67.5–76.5)
Absent/sporadic FM and abnormal imaging *FN = 16, FP = 21*	61.9 (45.6–76.4)	94.2 (91.3–96.4)	55.3 (43.4–66.6)	95.5 (93.5–96.9)	90.8 (87.6–93.5)
Absent FM and/or abnormal imaging *FN = 5, FP = 81*	88.1 (74.4–96.0)	77.6 (72.9–81.8)	31.4 (26.8–36.3)	98.3 (96.1–99.2)	78.7 (74.3–82.6)
Absent FM and abnormal imaging *FN = 16, FP = 3*	61.9 (45.6–76.4)	99.2 (97.6–99.8)	89.7 (73.3–96.5)	95.7 (93.8–97.1)	95.3 (92.7–97.1)

PPV, positive predictive value; NPV, negative predictive value; FMs, fidgety movements; FN, false negatives; FP, false positives.

**Table 3 jcm-08-01790-t003:** Risk factors and later cerebral palsy. The effects of the General Movements Assessment (GMA) during the fidgety movements period (FM) period, neonatal imaging, gestational age, and sex on the odds of developing cerebral palsy in a cohort of 405 high-risk infants.

	Cerebral Palsy			
	Unadjusted		Adjusted ^1^	
	OR (95 % CI)	*p Value*	OR (95 % CI)	*p Value*
Absent/sporadic FMs	15.3 (7.0–32.0)	<0.001	13.9 (5.6–34.6)	<0.001
Abnormal neonatal imaging	24.7 (10.8–56.3)	<0.001	29.3 (10.8–79.5)	<0.001
Gestational age Male sex			0.93 (0.87–1.0) 1.8 (0.8–4.6)	0.067 0.17

Binomial logistic regression. ^1^ adjusted for gestational age and sex. OR, odds ratio; CI, confidence interval.

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
