# Peer review of "The Predictive Accuracy of the General Movement Assessment for Cerebral Palsy: A Prospective, Observational Study of High-Risk Infants in a Clinical Follow-Up Setting"

_jcm, 2019, doi:10.3390/jcm8111790_

Round 1

Reviewer 1 Report

General Comments:

Good study with a large sample across multiple sites This is a worthwhile addition to the published information on use of GMs in clinical settings Interesting as this study adds information to the prediction of sporadic fidgety movements

Amendments needed:

Introduction:

Line 82: You state that – “Additionally, high sensitivities and specificities are often reported by GMA experts who are also tutors [12-16]”

Reference 16 (Morgan et al, 2016) – this study was not scored by any GMs tutors. It was conducted across multiple clinical settings and scored by experienced GMs trained clinicians (none of whom were GMs tutors)

Results:

“Infants with video recordings which could not be assessed with GMA due to inadequate video set-up (n=5), wrong state of the infant (n=8), restricted movements due traumatic injury (n=1) or external distractions to the infant (n=2), or who had significant medical incidences occurring after GMA assessment (n=3), or were categorized as having exaggerated FM (n=7), and children who were lost to follow-up (n=19) were excluded, leaving 405 infants available for analysis”.

This sentence is too long. Also, please consider re-wording “wrong state of the infant”

Discussion:

Comment 1

You argue “that the high accuracy of GMA demonstrated in highly skilled academic settings may not be reproducible in an ordinary clinical setting” and that “Many of the published studies on GMA have been conducted by researchers and GMA tutors”

This is a valid point however it’s not clear whether the GMs videos in your study were scored within ordinary clinical settings either. The paper states that 2 GMA observers with long experience and demonstrated reliability scored all videos (as opposed to the clinicians in each setting).

Comment 2

Line 242. You state that “Infants with severe congenital heart disease who needed early surgery and infants with complex medical conditions requiring prolong stay in the NICU are rarely included in studies on GMA…”

Please consider referring to the large Australian study on GMs in 278 infants following surgery, which included 149 infants with severe congenital heart disease. This study had low prevalence of CP but strong prediction for CP.

Crowle C, Galea C, Walker K, Novak I, Badawi N. Prediction of neurodevelopment at one year of age using the General Movements assessment in the neonatal surgical population. Early Hum Dev. 2018; 118:42-7.

Comment 3

Regarding sporadic fidgety movements. You state that although very few infants with sporadic fidgety GMs developed CP, other neurodevelopmental delays cannot be excluded

This is a good point but shoul dbe expanded. Please also comment on the other half of the infants with absent fidgety GMs who did not develop CP – did they have other neurodevelopmental delays or motor problems? This is important in the context of how ‘well performing’ the GMs assessment is.

Comment 4

In Limitations section (first paragraph definitely needs modifying): You state that “Longer follow-up might also have resulted in more children presenting with a mild CP phenotype not identified during the study follow-up” – Yes, agree. “In accordance with the strong association between presence of FM and milder types of CP, we speculate that this could have resulted in more infants with present FM being diagnosed with CP and, consequently, poorer performance of GMA”.

This is confusing. It sounds like you are saying that “normal fidgety GMs” are associated with mild CP (which is not correct). Please clarify this. It would be clearer to say ‘presence of sporadic fidgety movements’ (not “presence of FM”), as only 4% of those with intermittent fidgety GMs in your study were diagnosed with CP; and none of the infants with continuous fidgety movements.

It is also important to state that with longer term follow-up it is just as likely for infants with either absent or sporadic fidgety movements also later being diagnosed with a milder form of CP – and this would then show better performance of the GMA. I understand your point here but it I think it should be a more balanced argument. 

Reviewer 2 Report

This study used ratings on the General Movement Assessment (GMA) to predict development of cerebral palsy later in life in a large group of high-risk infants. Certified raters used only one rating session on which to base their decisions to more closely simulate clinic settings. Results show that GMA rating combined with imaging was most predictive of CP development but that sensitivity and specificity was lower compared to previously reported results. Authors interpret this to be a result of using a broad sample of high-risk infants rather than a restricted sample based only on those with abnormal imaging, for example. This study appears to provide a more realistic inclusion criteria, more representative of a wider population of at-risk infants which is an important contribution to the field. Imaging: CT vs MRI sensitivity – some discussion could focus on the relative sensitivity of the imaging modality. MRI seems more sensitive than CT for detecting damage in neonates. This may have affected the specificity and sensitivity of this category.

Author Response

Please see attachment (cover letter).

This manuscript is a resubmission of an earlier submission. The following is a list of the peer review reports and author responses from that submission.

Round 1

Reviewer 1 Report

Thank you for the opportunity to read this interesting manuscript. It concerns prediction of cerebral palsy (CP), using General Movements (GM). The early recognition of CP is of great interest, opening possible therapeutic windows. Five site were involved, recruiting high-risk infants for follow-up, GM was done once. These sites had different inclusion criteria and follow-up which makes this a challenge  to coordinate and summarize data.

Abstract: Good summary. Follow-up time is missing.

Introduction: It is stated that most children have brain lesions of perinatal origin - this is true for the high risk infants you are dealing with here, but only about 60-70% of children who later are diagnosed with CP are ever recognized as risk children and/or receive neonatal care. This statement should be modified and referenced.

Material and Methods: The criteria for imaging and the classifications used seem relevant. What was the minimum follow-up time? I may have missed it? Did it differ across sites?

How was the regression made, it would be interesting to see a more detailed description.

Results: Illustrates the challenges of summarizing data of different origin and collected for different purposes, but also the clinical perspective. It is logical that children with absent FM had severe impairment, but in the total group there may be more children with CP, not yet discovered! Moreover, the CP subtype is also recommended to be decided at 4-5 years of age, thus, the described distribution may change.

The lower sensitivity/specificity of prediction can maybe be expected in this less "selected" group.

Discussion: A thorough account of the results in relation to current literature and of the limitations of this study. This is a study connecting previous results to clinical reality in a way, where the ideal circumstances are not there. It should be underlined that follow-up time should be considered to be long(er)and that not all children are identified high risk infants.